# Comparative effects of French Contrast Method vs. Complex Training on explosive power and its endurance in youth badminton athletes

Ruiyin Huang[1], Yuhua Gao[1]*, Ke Yang[2], Yong Mo[1], Yongren Lu[3], Zhan Gao[4]*

**1** Guangzhou Sport University, Guangzhou, Guangdong, China, **2** Purple Tea Middle School, Jiangmen, Guangdong, China, **3** Youth Competitive Sports School, Guangdong Provincial Sports Bureau, Guangzhou, Guangdong, China, **4** Capital University of Physical Education and Sports, Beijing, China

* gaoyh@gzsport.edu.cn (YG); gaozhan@cupes.edu.cn (ZG)

## Abstract

Badminton players normally prioritize technical and tactical training over physical conditioning before competition, presenting a challenge in enhancing physical fitness within a constrained timeframe. While evidence have indicated complex contrast training can enhance strength and power, it is still unclear whether the French Contrast Method Training (FCMT) can bring greater explosive power gains to young badminton players. This study evaluated the effects of French Contrast Method Training versus Complex Training on lower limb explosive strength and its endurance in elite adolescent male badminton players. In a single-blind randomized controlled trial, 20 athletes were allocated to either an FCMT group (n = 10) or a CT group (n = 10) and completed an 8-week intervention. Performance was assessed pre- and post-intervention using standardized tests, including countermovement jump (CMJ), squat jump (SJ), drop jump (DJ), 10-m sprint, 5-0-5 agility test, eccentric utilization ratio (EUR), one-repetition maximum (1RM) squat, and lower extremity explosive endurance (LEEE). Subjective fatigue was monitored using the rating of perceived exertion (RPE) scale. Results demonstrated that while both training modalities improved explosive strength, the French Contrast Method Training led to significantly greater improvements in CMJ, DJ, Reaction Strength Index (RSI), EUR, agility, and LEEE (p < 0.05). The French Contrast Method Training also induced a larger effect size across most performance indicators. In contrast, the Complex Training improved strength and speed-related measures but showed limited effects on stretch-contraction cycle (SSC) utilization and endurance. Perceived fatigue was consistently higher in the CT group than in the FCMT group. These findings suggest that adaptation occurred more rapidly in the CT group (after roughly 3 weeks), whereas the FCMT group showed a more prolonged adaptation period (~1 month). The findings suggest that the French Contrast Method Training is more effective than the Complex Training in developing reactive strength, agility, SSC efficiency, and explosive

**Data availability statement:** The data that support the findings of this study are available in Figshare at https://doi.org/10.6084/m9.figshare.30349558.v1.

**Funding:** The author(s) received no specific funding for this work.

**Competing interests:** The authors have declared that no competing interests exist.

endurance in youth badminton athletes within a short-term training period. Given its neuromechanical advantages across the force–velocity spectrum, the French Contrast Method Training is recommended for use in the pre-competition phase of training, while Complex Training may be more suitable for foundational strength development. These results provide practical insights for coaches and support the strategic integration of French Contrast Method Training into performance enhancement programs for adolescent athletes.

## Introduction

Strength is a fundamental quality of the human body for sports, essential for developing other physical attributes and improving sports performance and competitive outcomes [1]. Many researchers are dedicated to discovering advanced training methods to improve athletes' muscle strength and performance [2,3]. Badminton is a multidirectional explosiveness-dependent sport [4], and the key to success lies in showing intense rhythmic movements, which include shuffling, jumping, twisting, stretching, and striking combined with a superior reactive ability [5]. A recent study quantifying muscle synergies during high-velocity badminton techniques revealed that performance is dependent on highly specialized neuromuscular coordination patterns to achieve maximal movement velocity [6]. This complex neuromuscular control mechanism indicates that explosive power in badminton stems not only from muscle strength and power output, but also requires the result of efficient and coordinated muscle recruitment. However, these qualities possess distinct physiological adaptive demands, making their simultaneous development within a training program a complex task for practitioners.

The Post-activation performance enhancement (PAPE) is a physiological phenomenon characterized by a transient increase in muscular performance following a conditioning activity, serving as the foundational mechanism for a variety of contemporary training strategies [7]. One such strategy is Complex Training (CT), which alternates high-intensity strength exercises with biomechanically similar plyometric or ballistic movements within a single session [8–10]. Complex training takes advantage of the PAPE mechanism to efficiently and simultaneously develop strength and explosive power in a single training session, meeting the training needs of practitioners. A more structured variant is the French Contrast Method Training (FCMT). Developed by French track and field coach Gilles Cometti and later promoted by Dietz. The FCMT employs a precise sequence of four exercises (large load→bodyweight jumps→light load→assisted jumps) with varying loads to maximize PAPE, thereby enhancing short-term explosive performance (e.g., in jumping and sprinting) and bringing about a significant anaerobic challenge that improves physiological endurance [11]. It is theorized that this approach may also foster long-term adaptations for more efficient power production [11].

Recent updates to badminton rules, techniques, and tactics have introduced a new level of unpredictability to match outcomes, challenging players to adapt to their

bodies in new ways. Chunlei Li's analysis highlights the critical role of short bursts of explosive power in matches where the ratio of net playing time to intervals is approximately 1:2, with rounds lasting less than 10 seconds accounting for 80% of matches [12]. However, with the shift in modern tactics to longer rounds of more than 10 seconds, with short breaks between scores (typically 27–30 seconds), the average length of the game has become longer, implying that sustaining explosive endurance is important to winning the game [12,13]. During long matches, players must demonstrate the explosiveness and endurance to handle frequent starts, stops, and quick movements in limited space. This requires strong muscle contractions, as well as effective utilization of the stretch-shortening cycle. Previous studies have shown that combining resistance training and plyometric training can effectively enhance the agility and reactive strength of athletes [14]. However, the impact of FMCT on the agility and reactive strength of athletes remains unclear.

Badminton players traditionally prioritize technical and tactical training over physical conditioning before matches, presenting a challenge in enhancing physical fitness within a constrained timeframe. Evidence indicates that CT's appeal among coaches stems primarily from its time efficiency, as this method can yield comparable improvements in athletic performance to traditional strength and power training, but in a more time-efficient manner [15,16]. Several studies have explored the short- or long-term effects of FCMT on adult players on various outcome indicator [17–22]; however, there is limited research on the long-term effects of FCMT on adolescent badminton players. In addition, few studies have compared FCMT and CT with each other and few studies have explored the adaptation period of athletes to FCMT. Therefore, this study therefore aimed to determine if 8 weeks of French Contrast Method Training (FCMT) is more effective than Complex Training (CT) at improving lower limb power and its endurance in junior badminton players, while also evaluating the adaptive responses to both training modalities.

## Methods

### Study design

The research experiment was a two-arm, single-blind, randomized controlled trial in which the subjects were not aware of the purpose of the intervention or the grouping, and the design and methodology were conducted in strict accordance with the requirements of the Ethics Committee of Guangzhou Sports Institute. All subjects signed a written informed consent form; for minor participants, their guardians also signed the consent form. The recruitment period for this study is from May 5, 2023 to May 6, 2023. The study was reviewed by the Ethics Committee of Guangzhou Sports Institute (No. 2023LcLL-14; date: May 4, 2023) and was registered with the China Clinical Trial Registry (registration number: ChiCTR2400092439). All subjects were familiarized with both interventions and testing procedures before the intervention period. Subjects who met the inclusion criteria were randomized into two groups: the FCMT group and the CT training group. The random assignment method employed a simple random draw: 1. Participants were ranked from 1 to 10 based on the pre-test list order. The pre-test list was sorted in ascending order by the number of strokes in surnames. If the number of strokes in surnames was identical, the third character of the full name was used for ascending order. No participants in this experiment required comparison of the second character in their surnames. 2. Prepare 10 distinct slips of paper numbered 1–10, crumple them into balls, place them in a sealed box, shake thoroughly, and draw the slips. 3. Starting from the first draw, odd-numbered slips were assigned to the FCMT group (the intervention group), while even-numbered slips were assigned to the CT group (the control group). Except for the pre-test list sorting performed by the author, all randomization procedures were conducted by an uninformed third party. All subjects completed five baseline tests before randomization into groups, including jump tests (CMJ, SJ, and DJ), 10-meter sprint test, agility test (5-0-5 test), Lower Extremity Explosive Endurance Test, and Maximum Strength Test, and the EUR was calculated based on the percentage of the jump test, which EUR, Agility Test, 10m Sprint Test, and Maximum Strength Test were used as the primary metrics for assessing lower extremity explosive endurance, and the jump test metrics were used as secondary metrics in this study and for variability analysis of specific gains. To isolate the effects of the two distinct training

structures and observe the resulting adaptation patterns without the confounding influence of load progression, a fixed training load was deliberately employed for both groups throughout the 8-week intervention period. Both groups therefore followed an identical loading scheme (load weight, number of repetitions, and number of sets) during all sessions. The training interventions and tests were conducted in the physical training hall of the Guangdong Youth Sports Competition School. Since the participants in the experiment were all from full-time sports training institutions, their specialized training arrangements, dietary habits, and work and rest schedules were uniform to minimize the influence of external factors on the experimental results.

## Participants

Sample needs were estimated through a pre-test power analysis (G*Power 3.1) (modeled as a paired t-test, $\alpha = 0.05$, power = 0.80, expected effect size = 0.80), which indicated a need for 24 participants, but was limited by the scarcity of this group given that the study population was elite youth athletes (all provincial or national youth games medalists), seasonal scheduling and training consistency requirements, 20 athletes were ultimately included (FCMT = 10; CT = 10). The situation where each group has a sample size of n ≤ 10 has been observed in previous similar studies [17,18,23]. In addition, in order to avoid unnecessary injuries and interference caused by asymmetry or safety hazards in squatting movement patterns, overhead squatting (one of the functional movement screens) was used as a subject screening index (those with FMS scores <1 were not included). As shown in Fig 1, 20 adolescent male badminton players volunteered to participate in this study and were randomly divided into the FCMT group (n = 10) and the CT group (n = 10) with the following inclusion criteria: 1) no impairment in squatting movement patterns (over-the-shoulder squat score of at least 2), 2) at least three years of resistance training experience, 3) participation in provincial/national youth tournaments, and 4) no history of physical or mental injuries from the beginning of the study to the end of the study there was no history of physical or mental injury. Participants who did not meet these criteria were excluded from the experimental results. Eligible participants completed the full intervention training and testing sessions. In addition, considering that physiological maturity can affect the training response of intermittent training [24], and then this study assessed differences in physiological maturity of subgroups by testing the age at peak height velocity (APHV) [25] of the subjects to exclude interference from ongoing growth spurts at this age.

## Training program

The 8-week intervention comprised two supervised sessions per week (Mondays, 14:00–15:30; Fridays, 16:00–17:30) separated by closer to 72 hours minimum. Each session began with a standardized 20-min dynamic warm-up (mobility drills, dynamic stretching, activation) and concluded with a 10-min cool-down (static stretching). To ensure safety and technical consistency, two certified coaches supervised each station, provided spotting for barbell exercises, and reinforced proper technique. All coaches were trained on the study protocols before the intervention.

Before the intervention, participants completed a familiarization phase consisting of coach-led demonstrations and light-load practice of all exercises from both protocols to standardize technique and execution intent. The French Contrast Method Training (FCMT) and Complex Training (CT) protocols were co-designed by the investigators and certified strength and conditioning specialists to ensure fidelity to their respective theoretical frameworks. The interventions differed primarily in (i) organizational structure and (ii) PAPE activation strategy. The FCMT and CT training protocols were jointly developed by researchers and certified strength and conditioning specialists to ensure adherence to their respective theoretical frameworks (Table 1). The protocols differ fundamentally in two aspects: organizational structure and PAPE activation methodology. In terms of organizational structure, FCMT employs a sequential four-movement framework adapted from Hernández et al. [19], consisting of: high-intensity movement (isometric back squat), augmentation movement (drop jump), light-loaded movement (back squat jump), and accelerated augmentation movement (band-assisted jump). In contrast, CT follows a paired-exercise structure with two distinct cycles: Cycle 1 (back squat + drop jump) and Cycle 2 (back

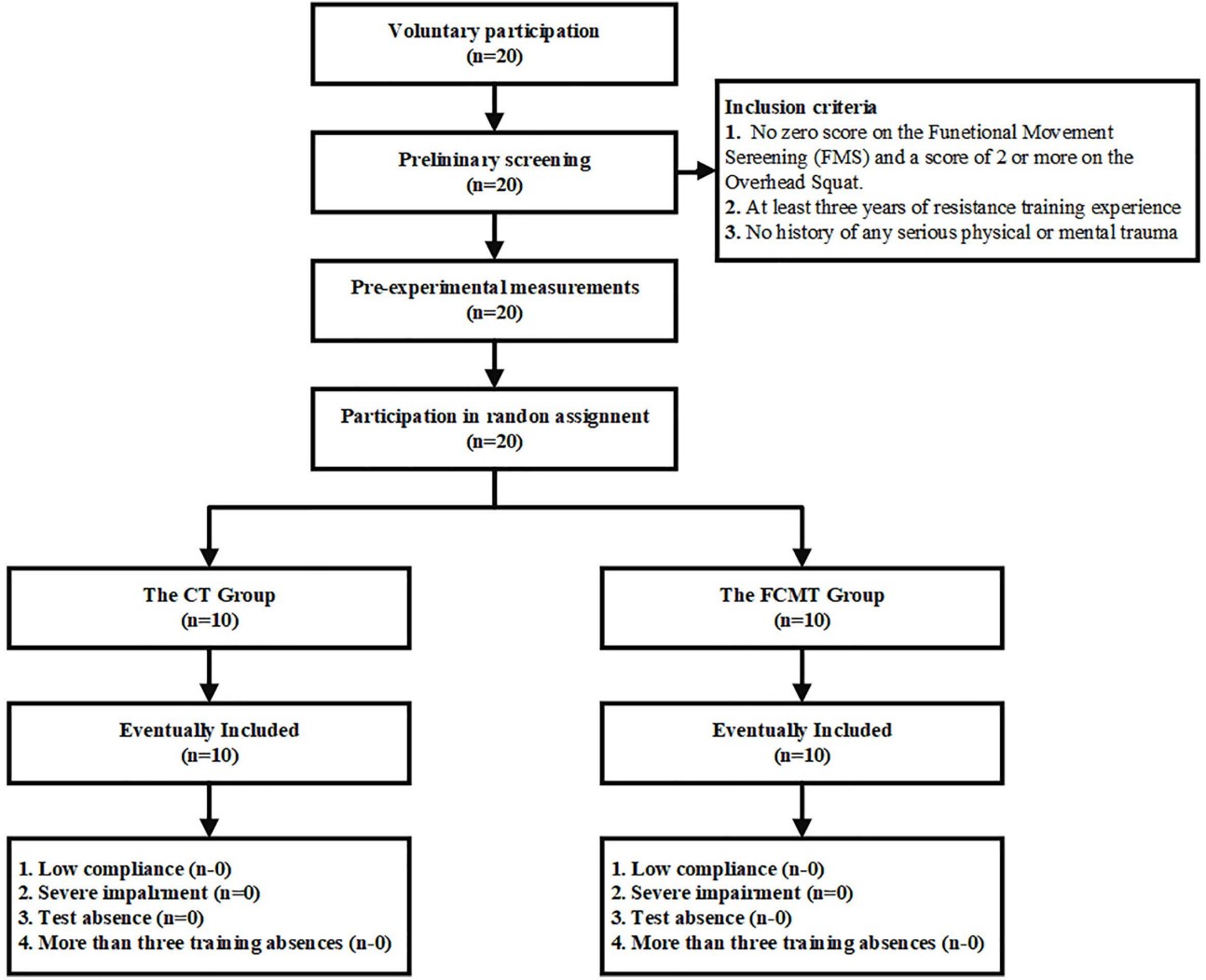

**Fig 1. Participant screening flowchart.**

**Table 1. Training arrangement of the FCMT group and the CT group.**

| Group | Movement for training | Load strength | Rep | Set | Intermission |
|---|---|---|---|---|---|
| The FCMT group | Isometric back squat | 85%1RM | 4 | 4 | 20s |
| | Drop jump | 50 cm | 4 | | 20s |
| | Back squat jump | 50%BW | 8 | | 20s |
| | Band-assisted jump | | 8 | | 3-4 min |
| The CT group | Back squat | 85%1RM | 4 | 4 | 20s |
| | Drop jump | 50 cm | 4 | | 3-4 min |
| | Back squat jump | 50%BW | 8 | 4 | 20s |
| | Single-leg box jump | | 8 | | 3-4 min |

Note: BW = body weight; Rep = repetition; Set = Number of repeated groups.

squat jump + single-leg box jump). Regarding PAPE activation, FCMT utilizes isometric back squats during the high-load phase based on evidence demonstrating the superior efficacy of isometric contractions for inducing PAPE [26–29], while CT employs back squats. This strategic difference reflects FCMT's emphasis on optimizing neural excitation through isometric loading compared to CT's focus on dynamic movement patterns. Both groups completed four sets per session with identical rest intervals (20-second intra-cycle and 3–4 minute inter-set recovery) and maintained consistent external loading parameters. The FCMT group (10 participants) and the CT group (10 participants) were divided into three groups for multiple-person circuit training (number of participants per group was distributed as follows: 2 groups × 3 participants/ group, 1 group × 4 participants/group). All movements were executed at maximal velocity with high concentration, except during high-load exercises. Prior to the actual intervention, participants were familiarized with the two training protocols by watching a demonstration by a professional fitness coach and performing a lighter-load training protocol.

**Testing program**

As shown in Fig 2, the study arranged two main assessments, a pre-test (T1) before randomization and a post-test (T2) after an 8-week training period, where participants were given time to familiarize themselves with the test program during the week prior to testing. Considering the mutual exclusion of energy expenditure between the test program sequences, based on the testing principle of explosive strength testing before and endurance or maximal strength after; the testing procedure was divided into three days during the testing week (Monday, Wednesday, and Friday), with the 10-m Sprint, the CMJ test, and the SJ test performed on Monday, the body composition measurements and the Lower Extremity Explosive Endurance Test on Wednesday, and a 5-0-5 agility test, DJ test, and Maximum Strength Test on Friday. The T1 and T2 tests were always uniform in terms of test sequences, subject, and tester organization. To ensure reliable results, subjects maintained a consistent diet before testing, got adequate sleep, and avoided strenuous physical activity.

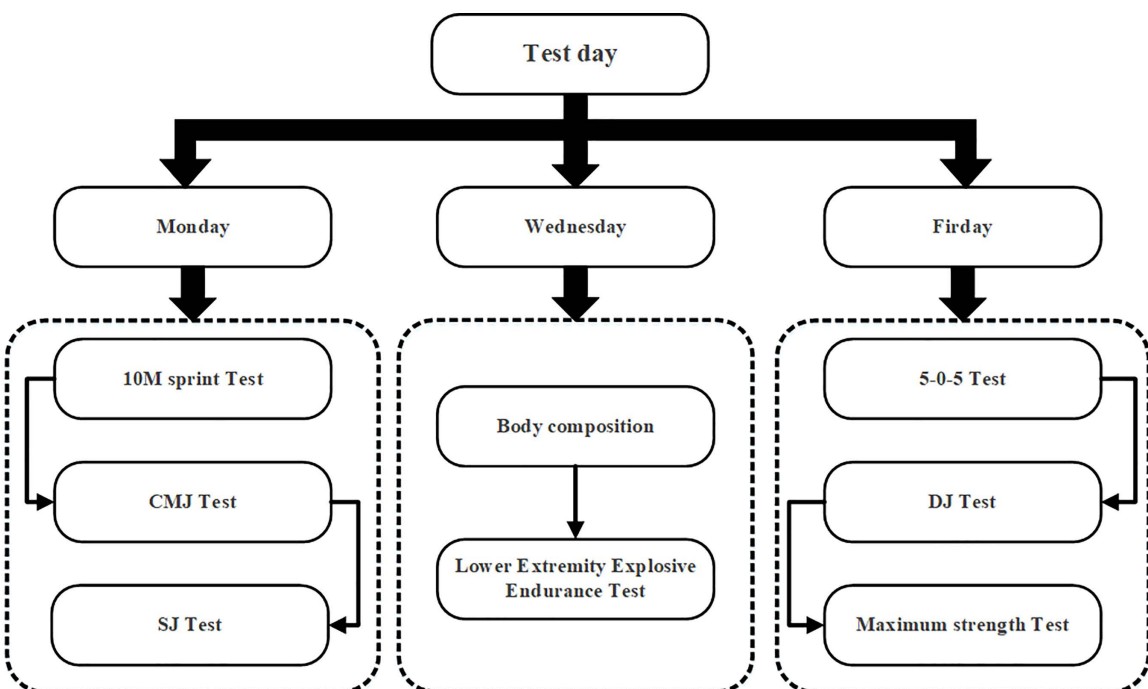

**Fig 2. Test flow arrangement diagram.**

## Test procedures

**Body composition.** The body composition of participants was assessed using a bioelectrical impedance analyzer (Inbody), both before and after the experiment. Measurements of body height and mass were taken, from which the Body Mass Index (BMI) was calculated. To ensure accuracy, participants were asked to refrain from eating or drinking for 4 hours before the test and the subsequent warm-up. Additionally, the hip joint served as the reference point for measuring leg length and sitting height, critical for calculating the APHV [25].

**Vertical longitudinal jump test.** Vertical jump tests consisted of the Counter movement jump (CMJ), Drop Jump (DJ), and Squat Jump (SJ). The tests were conducted using the Australian Smart Jump wireless test mat, Fusion Sports Smart Jump Mat, and each jump test was performed twice with a minimum of 1 minute between each test, and the average of the 2 tests was used as the basis for data analysis, with jump height and relative peak power in the jump tests collected as outcome indicators (in the case of the DJ test, the Reaction Strength Index (RSI) was included as one of the outcome indicators). In addition, the intra-group correlation coefficients of the pre-intervention tests showed good reliability and reproducibility (0.97, 0.92, and 0.91 for CMJ, SJ, and DJ heights, respectively; 0.97, 0.93, and 0.91 for relative power, respectively; and 0.82 for DJ-RSI).

First, the CMJ was used as an indirect measure of an athlete's explosive lower body strength [30,31]. Participants were asked to stand at the center of the jumping mat, place their hands on their hips, squat at a controlled speed to an optimal depth, then jump upward with full force and land on the mat as close to the initial position as possible after reaching maximum height.

Second, the SJ test is often used to measure an athlete's lower body explosive power (speed strength ability) [30,31]. The participant remains stationary in a semi-squat position for 3 seconds and then starts jumping upwards. The arm position and body position during the test are consistent with the CMJ test.

Finally, the DJ test is used to assess an athlete's reactive jumping ability and explosive power in dynamic vertical jumps [30,31]. Participants are asked to place their hands on their hips, free-fall from a platform 40 cm above the ground, jump upwards with maximum effort to minimize time in contact with the ground, and then free-fall to the jumping mat after reaching maximum height.

**Eccentric Utilization Ratio (EUR).** EUR is used to assess an athlete's ability to utilize the stretch-contraction cycle (SSC) (A biomechanical mechanism in which a muscle contracts rapidly after being passively stretched, triggered by a neural reflex.), which is important in many sports [32,33]. EUR is the difference between CMJ height and SJ height, which reflects the efficiency of the SSC.

**Sprint test and agility test.** The Sprint test was used to assess athletes' linear speed and acceleration, key factors reflecting explosive lower limb strength [30,34]. Participants were tested on a 10-meter elapsed time using the Smart Speed Fusion exercise. Subjects were asked to start standing in front of the starting line (0.5 M from the first lighted gate) and then decelerate and cushion after passing through the second lighted gate as hard as they could. Each participant made 2 attempts with a minimum of 3 minutes between attempts. The 10M sprint test was performed 2 times with good reliability and reproducibility for ICC (ICC: 0.74).

The 5-0-5 test was used to evaluate the tester's ability to change the direction of movement quickly, and its reliability has been confirmed in previous studies [35,36], and it has been proved that there is a high correlation between the reactive strength and the Change of Direction (COD), and that the body changing direction involves a relatively small angle of knee flexion and extension and a shorter ground contact time [37], which is conducive to the enhancement of the badminton players' efficiency in attacking, defending and stroke quality. A 15-meter-long flat and open running track was set up, with a starting line and a marking line at each end (the width of both was not included in the length of the track), and timing gates were placed 5 meters away from the marking line (on both sides of the track). Subjects from the starting line with maximum effort sprint to the marking line, must touch the line with one foot and complete a 180 ° turn (not touching the ground with their hands), and then return to the starting line at full speed. The optimal time for the 5-meter round trip

(i.e., from touching the line to the 5-meter timing gate) was recorded as the test data, and the test was performed twice with a 2-minute rest period between each test, and The ICC had good reliability and reproducibility (ICC: 0.71).

**Lower Extremity Explosive Endurance Test (LEEE).** The LEEE protocol used in this study was adapted from the intermittent vertical-jump paradigm of Hespanhol et al., originally developed for volleyball, to better reflect badminton's repeated, short-interval, high-intensity demands [38]. External load was set at 30% 1RM, following Stone et al., who identified ~10–40% 1RM as optimal for maximizing lower-body power output, thereby supporting the test's validity for explosive-strength qualities [39]. The set comprised 15 repetitions, a volume selected to elicit appreciable fatigue for endurance assessment while minimizing technical degradation, consistent with prior repeated-jump protocols [38]. Procedurally, participants performed 15 continuous loaded squat jumps at maximal velocity with standardized technique cues (hands on bar, full foot contact, consistent countermovement depth). Repetitions were paced by an audible metronome (researcher-controlled) to avoid clustering and ensure uniform inter-rep timing. Verbal encouragement was standardized, and safety spotting was provided throughout. Given the high physiological load, no retesting was scheduled. To enhance reliability, all participants completed a formal familiarization session one week prior to testing that included instruction, equipment setup, and practice trials at the prescribed load. Evidence indicates that jump tests conducted with comparable instrumentation and procedures exhibit excellent reliability, providing indirect support for the present protocol [40].

**Maximum strength test.** There are various methods for assessing the maximum strength of athletes, including predictive models and direct measurement method [41–43]. This study employed direct measurement method for testing. At the end of a standardized warm-up routine, subjects performed a series of weighted deep squats consisting of: 50% 1RM (5 reps), 70% 1RM (3 reps), 80% (2 reps), and 90% (1 rep). Finally, a separate 1RM test was performed, each time the weight was increased by 4 kg or decreased by 2 kg. A 3- to 5-minute rest period was allowed between all tests, and all participants were asked to reach the 1RM in 3–7 attempts. Participants were verbally encouraged and observed for depth of squat (thighs parallel to the floor). The final successful squat weight was recorded as the 1RM weight. In the direct measurement method, the participant increases the weight incrementally, and in order to ensure that the participant is exerting maximum effort in each squat, the maximum strength test is performed only once, therefore no ICC calculation is performed.

**Monitoring indicators and processes.** Exercise-induced fatigue, the phenomenon of a temporary decrease in physical function due to workload, is both an expected outcome and a stimulus during adaptation. In order to track athletes' fatigue and adaptation during training, the study utilized the subjective Rating of Perceived Exertion (RPE) scale [44,45]. Subjects were asked to complete a self-perceived fatigue assessment scale for fatigue monitoring after each training session. In addition, subjects completed a familiarization with the RPE scale during the screening phase of the experiment to ensure that each fatigue monitoring data formally included was not adversely affected by cognitive differences.

## Statistical analyses

In this study, we analyzed basic participant characteristics including age, training years, height, weight, BMI, Age at peak height velocity (APHV), and Over-the-shoulder squat score. Outcome measures included sprint tests (10M sprint, 5-0-5 test), CMJ and SJ (heights and relative power), DJ performance (height, relative power, Reaction Strength Index), SSC performance indices (EUR), 1RM deep squat performance, and LEEE results. For descriptive statistics, we reported Means and Standard Deviations (SD) for normally distributed data, and medians along with range (maximum, minimum) for non-normally distributed data. Normality was checked using the Shapiro-Wilk test. Reliability was assessed using the Intraclass Correlation Coefficient (ICC), with values above 0.7 indicating high reliability [46]. However, due to the physically demanding nature of the Maximum Strength and Lower Explosive Endurance Tests, these were not repeated and thus not included in the reliability assessment. For inferential statistics, non-normal distributions or variances prompted the

use of the Mann-Whitney U test or the Wilcoxon test to analyze differences within and between groups. In cases of normal distribution, independent samples t-tests were used to examine baseline differences between groups, and paired samples t-tests were employed to compare changes before and after training within groups. For the pre- and post-intervention comparisons between the two groups, a two-way mixed-effects ANOVA was employed to examine the independent effects of group (FCMT vs. CT) and time (pre vs. post) on each dependent variable, and post hoc comparisons were performed using Tukey's method. If data did not meet normality assumptions, nonparametric analysis (Kruskal-Wallis test) was performed, with post hoc comparisons conducted using the Dwass-Steel-Critchlow-Fligner method. A p-value of less than 0.05 was considered statistically significant. Given the small sample size, Hedges' g value was used as the effect size to compensate for the statistical power. Effect size (ES) was calculated to assess the magnitude of changes, with Hedges' guidelines [47] indicating small (0.2), medium (0.5), and large (0.8) effects for normally distributed data. For non-normally distributed data, the Rank Biserial Correlation (RBC) and Cureton's guidelines were applied [48]. All statistical analyses were conducted using the jamovi software package (jamovi 2.2.2).

## Results

As shown in Fig 1, no adverse reactions due to injury or early study withdrawal were reported by any subject during the 8-week study period. Furthermore, based on our observations, all subjects scored 3 points in the overhead squat test. Ultimately, data from 20 subjects (10 in the FCMT group and 10 in the CT group) were included in the statistical analysis. At baseline, there were no significant differences in descriptive variables or exercise parameters between the FCMT and CT groups (Tables 2 and 3).

### Fatigue monitoring metrics results

As shown in Fig 3, there was a general decreasing trend in the RPE index for both groups. The CT group consistently exhibited a higher level of perceived fatigue after the training session compared to the FCMT group. Notably, the CT group showed a greater increase in perceived fatigue than the FCMT group during the 6th training session. However, between the 6th and 9th training sessions, the CT group showed a significant decrease in perceived fatigue that exceeded the decrease observed in the FCMT. After the 9th training session, perceived fatigue levels decreased dramatically in the FCMT group by a greater magnitude than in the CT group. Experimental data indicate that perceived fatigue in the CT group showed a significant decrease by Week 3, whereas perceived fatigue in the FCMT group exhibited a significant decrease after one month. Therefore, the CT group appears to have adapted to the training demands earlier than the FCMT group.

Table 2. Table of basic characteristics of experimental participants.

| Characteristics | FCMT group | CT group | Normality test | P-values | SMD |
|---|---|---|---|---|---|
| Age | 15±1.15 | 14.5±0.972 | 0.76 | 0.31 | 0.45 (−0.45, 1.46) |
| Height (cm) | 171.2±8.82 | 171.7±7.565 | 0.11 | 0.90 | −0.05 (−1.00, 0.88) |
| Weight (kg) | 63.5±11.11 | 61.1±12.013 | 0.82 | 0.65 | 0.20 (−0.72, 1.17) |
| BMI | 21.5±2.56 | 20.6±3.312 | 0.53 | 0.50 | 0.29 (0.61, 1.28) |
| Years of Training | 7.0±0.82 | 6.20±1.87 | 0.32 | 0.23 | 0.53 (−0.21, 1.37) |
| APHV | 2.74±1.15 | 2.40±0.92 | 0.51 | 0.47 | 0.32 (−0.43, 1.13) |

Note: Values are expressed as median (range) or mean (SD). The p-value in the t-test or Mann–Whitney U test denotes the between-group difference between the Experimental and Control groups; * represents a significant difference between the groups, and SMD stands for standardized mean difference.

**Table 3. Statistical table of the differences between the indicators measured before the experiment in the two groups.**

| Test Indicator | CT group | FCMT group | Normality test | P-values | SMD | ICC |
|---|---|---|---|---|---|---|
| T-10M (s) | 2.00±0.01 | 1.99±0.13 | 0.91 | 0.84 | −0.09 (−0.87, 0.68) | 0.74 |
| 505 (s) | 2.39±0.12 | 2.39±0.12 | 0.35 | 0.08 | −0.79 (−1.67, -0.06) | 0.71 |
| CMJ-Height (cm) | 39.67±6.09 | 38.97±5.08 | 0.49 | 0.78 | −0.12 (−0.91, 0.64) | 0.97 |
| CMJ-RP (W/kg) | 50.35±5.28 | 49.72±4.41 | 0.48 | 0.78 | −0.12 (−0.91, 0.64) | 0.97 |
| SJ-Height (cm) | 38.19±5.12 | 37.84±5.23 | 0.88 | 0.88 | −0.06 (−0.85, 0.71) | 0.92 |
| SJ-RP (W/kg) | 49.06±4.44 | 48.76±4.55 | 0.89 | 0.88 | −0.06 (−0.85, 0.71) | 0.93 |
| DJ-Height (cm) | 36.12±6.11 | 37.55±6.75 | 0.95 | 0.62 | 0.21 (−0.54, 1.01) | 0.91 |
| DJ- RP (W/kg) | 47.27±5.31 | 48.50±5.86 | 0.95 | 0.63 | 0.21 (−0.55, 1.01) | 0.91 |
| DJ-RSI | 1.00±0.28 | 0.96±0.16 | 0.67 | 0.70 | −0.17 (−0.96, 0.59) | 0.82 |
| EUR | 1.49±2.15 | 1.13±1.64 | 0.16 | 0.68 | −0.18 (−0.98, 0.58) | – |
| 1RM Squat (Kg) | 113.10±15.50 | 115.32±14.96 | 0.66 | 0.75 | 0.14 (−0.62, 0.93) | – |
| LEEE | 0.13 (0.09, 0.50) | 0.14 (0.11, 0.51) | <.001 | 0.47 | 0.10 (−0.66, 0.89) | – |

Note: Values are expressed as median (range) or mean (SD). The p-value in the t-test or Mann–Whitney U test denotes the between-group difference between the FCMT and CT groups; * represents a significant difference between the groups; RP relative power; SMD stands for standardized mean difference; ICC Intraclass Correlation Coefficient; LEEE Lower Extremity Explosive Endurance.

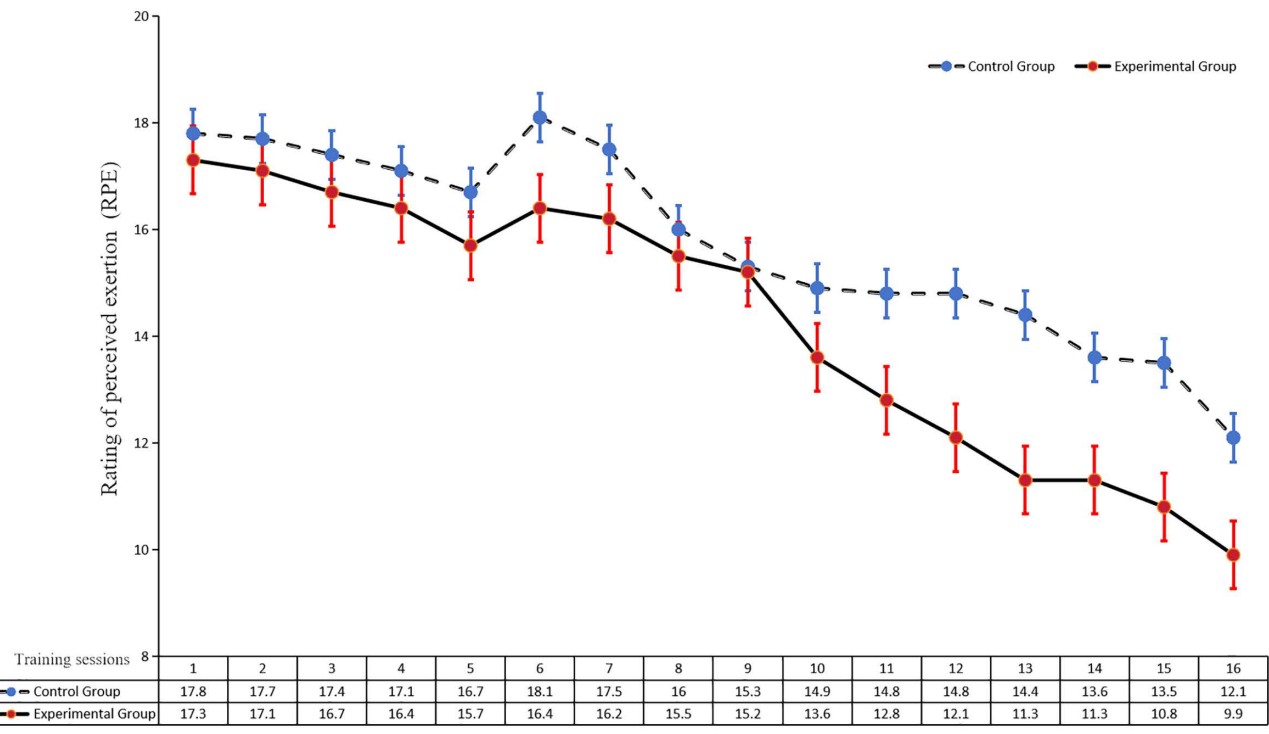

**Fig 3. Curve of mean change in RPE scale index after 16 training sessions in both groups.**

## Results of test data analysis

As shown in Tables 4 and 5, after 8 weeks of training intervention, all the exercise parameters in the FCMT group showed significant enhancement and high effect sizes, whereas in the CT group, except for DJ-RSI (p = 0.94, ES = 0.02), EUR (p = 0.70, ES = 0.11), and LEEE (p = 0.28, RBC = 0.40), the other exercise parameters also showed significant elevation and high effect sizes. This suggests that FCMT was able to significantly enhance overall explosive strength and explosive

**Table 4. Statistics of the results of the indicators before and after the experiment in the CT group.**

| Test Indicator | CT Group | | Normality test | P-values | ES/RBC | SMD (90%CI) | Change score (% Δ) |
|---|---|---|---|---|---|---|---|
| | Baseline | Post-intervention | | | | | |
| T-10m (s) | 2.00 ± 0.01 | 1.92 ± 0.14 | 0.92 | 0.01** | 1.02 | −0.70 (−1.13, -0.04) | −4% |
| 505 (s) | 2.46 (2.32, 2.65) | 2.44 (2.30, 2.55) | 0.04 | 0.01** | 0.91 | −0.46 (−0.81, -0.20) | −1.6% |
| CMJ-Height (cm) | 39.67 ± 6.087 | 43.01 ± 5.09 | 0.86 | <.01** | 1.91 | 0.57 (0.42, 0.83) | 8.4% |
| CMJ-RP(W/kg) | 50.35 ± 5.28 | 53.08 ± 4.35 | 0.90 | <.01** | 1.60 | 0.54 (0.38, 0.80) | 5.4% |
| SJ-Height (cm) | 38.19 ± 5.12 | 41.33 ± 4.34 | 0.49 | <.01** | 1.42 | 0.63 (0.41, 0.97) | 8.2% |
| SJ- RP(W/kg) | 49.05 ± 4.44 | 51.47 ± 3.79 | 0.58 | <.01** | 1.40 | 0.62 (0.40, 0.96) | 4.9% |
| DJ-Height (cm) | 36.12 ± 6.11 | 38.52 ± 5.33 | 0.18 | 0.01** | 1.25 | 0.40 (0.25, 0.62) | 6.6% |
| DJ-RP (W/kg) | 47.27 ± 5.31 | 49.06 ± 4.58 | 0.24 | 0.01** | 1.07 | 0.35 (0.20, 0.55) | 2.2% |
| DJ-RSI | 1.00 ± 0.28 | 1.01 ± 0.20 | 0.14 | 0.94 | 0.02 | 0.02 (−0.45, 0.50) | 1% |
| EUR | 1.49 ± 2.15 | 1.69 ± 1.44 | 0.67 | 0.70 | 0.11 | 0.11 (−0.33, 0.56) | 13.40% |
| 1RM Squat (kg) | 113.10 ± 15.50 | 125.90 ± 11.46 | 0.71 | <.01** | 2.57 | 0.90 (0.70, 1.25) | 11.3% |
| LEEE | 0.13 (0.09, 0.50) | 0.11 (0.09, 0.16) | <.01 | 0.28 | 0.40 | −0.54 (−1.53, 0.35) | −31.3% |

Note: ES = effect size (Hedges' g); RBC = Rank biserial correlation; CI = confidence interval; SMD = standardized mean difference; %Δ = (post-baseline)/baseline × 100%; * Significant difference before and after training (p < 0.05); ** Significant difference before and after training (p < 0.01). T-10m = 10-m short stroke time; LEEE Lower Extremity Explosive Endurance.

**Table 5. Statistics of the results of the indicators before and after the experiment in the FCMT group.**

| Test Indicator | FCMT Group | | Normality test | P-values | ES/RBC | SMD (90%CI) | Change score (% Δ) |
|---|---|---|---|---|---|---|---|
| | Baseline | Post-intervention | | | | | |
| T-10m (s) | 1.99 ± 0.13 | 1.85 ± 0.13 | 0.07 | <.01** | 2.12 | −1.04 (−1.53, -0.74) | −7.0% |
| 505 (s) | 2.39 ± 0.12 | 2.28 ± 0.11 | 0.7 | <.01** | 2.08 | −0.91 (−1.33, -0.65) | −4.6% |
| CMJ-Height (cm) | 38.97 ± 5.08 | 44.60 ± 5.95 | 0.23 | <.01** | 2.11 | 0.98 (0.70, 1.43) | 14.4% |
| CMJ-RP (W/kg) | 49.72 ± 4.41 | 54.27 ± 5.19 | 0.67 | <.01** | 1.82 | 0.90 (0.63, 1.34) | 9.2% |
| SJ-Height (cm) | 37.84 ± 5.23 | 41.62 ± 4.97 | 0.33 | <.01** | 2.28 | 0.71 (0.51, 1.04) | 10.9% |
| SJ- RP (W/kg) | 48.76 ± 4.55 | 51.72 ± 4.54 | 0.49 | <.01** | 1.78 | 0.62 (0.43, 0.94) | 6.1% |
| DJ-Height (cm) | 37.55 ± 6.75 | 41.64 ± 6.31 | 0.55 | <.01** | 2.40 | 0.60 (0.44, 0.87) | 10.9% |
| DJ- RP (W/kg) | 48.50 ± 5.86 | 51.96 ± 5.39 | 0.52 | <.01** | 2.45 | 0.59 (0.43, 0.85) | 7.1% |
| DJ-RSI | 1 (0.60, 1.20) | 1.25 (0.85, 1.85) | 0.05 | 0.01** | 1 | 1.27 (0.75, 2.02) | 31.3% |
| EUR | 1.13 ± 1.64 | 2.98 ± 2.31 | 0.74 | <.01** | 1.32 | 0.89 (0.58, 1.36) | 163.70% |
| 1RM Squat (Kg) | 115.32 ± 14.96 | 130.76 ± 12.23 | 0.94 | <.01** | 2.34 | 1.08 (0.80, 1.56) | 13.4% |
| LEEE | 0.14 (0.11, 0.51) | 0.08 (0.048, 0.090) | <.01 | 0.01** | 1 | −1.11 (−2.23, -0.20) | −52.9% |

Note: ES = effect size (Hedges' g); RBC = Rank biserial correlation; CI = confidence interval; SMD = standardized mean difference; %Δ = (post-baseline)/baseline × 100%. * Significant difference before and after training (p < 0.05); ** Significant difference before and after training (p < 0.01). T-10m = 10-m short stroke time; LEEE Lower Extremity Explosive Endurance.

endurance in adolescent athletes; CT was able to locally enhance explosive strength in adolescent athletes, but it was not significant for reactive strength, SSC utilization ability, and explosive endurance.

The results presented in Table 6 indicate that the FCMT group showed significantly greater gains in CMJ than the CT group ($P_{CMJ-H} = 0.02$, $ES_{CMJ-H} = 1.06$; $P_{CMJ-RP} = 0.04$, $ES_{CMJ-RP} = 0.89$), which suggests that the FCMT improved lower extremity explosive strength better than the CT. Similarly, in terms of reactive strength and agility, the FCMT group showed better gains than the CT group in both ($P_{DJ-H} = 0.03$, $ES_{DJ-H} = 0.98$; $P_{DJ-RP} = 0.02$, $ES_{DJ-RP} = 1.06$; $P_{DJ-RSI} < 0.01$, $ES_{DJ-RSI} = 1.21$; $P_{5-0-5} < 0.01$, $ES_{5-0-5} = 1.26$), which suggests that FCMT is more favorable to the reactive strength and agility abilities. In terms of SSC utilization capacity, the FCMT group was superior to the CT group ($P_{EUR} = 0.02$, $ES_{EUR} = 1.06$) and the CT group did not show significant improvement before and after the intervention ($P = 0.70$, $ES = 0.11$). In terms of explosive endurance, the FCMT group showed a better increase than the CT group ($P_{LEEE} = 0.03$, $RBC_{LEEE} = 1$); whereas the CT group did not show a significant change ($P = 0.28$), but showed a smaller medium effect ($RBC = 0.42$). This demonstrated that the FCMT significantly improved the athletes' explosive endurance, while the CT group showed a smaller effect on the athletes' explosive strength. However, in terms of maximal strength, both the FCMT and CT groups were effective in improving the athletes' maximal strength, and the difference in the increase before and after the intervention was not significantly different between the two ($P_{1RM\ Squat} = 0.27$, $ES_{1RM\ Squat} = 0.47$). Similarly, in terms of speed power, both the FCMT and CT groups were effective in enhancing athletes' speed power, both in longitudinal jumping (SJ) and horizontal acceleration (T-10m), but no significant differences were observed in the pre- and post-intervention differences in increases between the FCMT and CT groups ($P_{SJ-H} = 0.41$, $ES_{SJ-H} = 0.34$; $P_{SJ-RP} = 0.70$, $ES_{SJ-RP} = 0.16$; $P_{T-10M} = 0.07$, $ES_{T-10M} = 0.79$).

## Discussion

Explosive strength is influenced by multiple factors [49,50], so a single test to assess the effect of explosive strength enhancement has limitations, and the present study used a multidimensional explosive strength test to accurately analyze the mechanism of action of the two training methods. The results showed that the French contrast method training (FCMT) could significantly improve the lower limb explosive strength and explosive endurance of adolescent athletes, while Complex Training (CT) could significantly improve the lower limb explosive strength but not the explosive

Table 6. Statistical table of differences between the pre- and post-experimental means of the indicators in the two groups.

| Test Indicator | FCMT Group/ CT Group Mean Difference | Normality test | P-values | ES/RBC | SMD(90%CI) |
|---|---|---|---|---|---|
| T-10m (s) | 0.06 | 0.44 | 0.07 | 0.79 | 0.83 (0.10, 1.73) |
| 505 (s) | 0.07 | 0.4 | 0.01** | 1.26 | 1.32 (0.58, 2.32) |
| CMJ-Height (cm) | −2.29 | 0.2 | 0.02* | −1.06 | −1.11 (−2.06, -0.37) |
| CMJ-RP (W/kg) | −1.82 | 0.63 | 0.04* | 0.89 | −0.93 (−1.84, 0.20) |
| SJ-Height (cm) | −0.64 | 0.71 | 0.41 | −0.34 | −0.36 (−1.17, 0.39) |
| SJ- RP (W/kg) | −0.28 | 0.87 | 0.7 | −0.16 | −0.17 (−0.96, 0.59) |
| DJ-Height (cm) | −1.68 | 0.07 | 0.03* | −0.98 | −0.59 (−1.44, 0.15) |
| DJ- RP (W/kg) | −1.47 | 0.12 | 0.02* | −1.06 | −1.11 (−2.06, -0.38) |
| DJ-RSI | −0.29 | 0.55 | 0.01** | −1.21 | −1.27 (−2.25, -0.52) |
| EUR | −1.65 | 0.83 | 0.02* | −1.06 | −1.11 (−2.06, -0.37) |
| 1RM Squat (Kg) | −2.64 | 0.78 | 0.27 | −0.47 | −0.49 (−1.33, 0.25) |
| LEEE | 0.05 | <.001 | 0.03* | 0.58 | 0.35 (−0.40, 1.17) |

Note: ES = effect size (Hedges' g); RBC = Rank biserial correlation; CI = confidence interval; SMD = standardized mean difference; * Significant difference before and after training (p < 0.05); ** Significant difference before and after training (p < 0.01). T-10m = 10-m sprint time; LEEE Lower Extremity Explosive Endurance.

endurance. In the overall dimension, FCMT significantly outperformed CT in terms of explosive strength enhancement; in the local dimension, FCMT outperformed CT in terms of reactive strength, agility, and SSC utilization, while there was no significant difference between the two in terms of speed strength and maximal strength enhancement (Fig 4). Therefore, the explosive power improvement advantage of FCMT over CT mainly originated from the enhancement of reactive power, agility ability and SSC utilization ability.

The superior gains in reactive strength, agility, and stretch–shortening cycle (SSC) efficiency observed with French Contrast Method Training (FCMT) relative to Complex Training (CT) are plausibly attributable to differences in session organization and post-activation performance enhancement (PAPE) strategies. Whereas CT employs two independent contrast pairs, FCMT implements a sequential four-exercise progression that exposes the neuromuscular system to stepwise reductions in external load within each cycle. This configuration likely provides a broader stimulus across the force–velocity spectrum than paired contrasts. Conceptually, FCMT's multi-load sequence also accords with the contemporary joint-by-joint paradigm, emphasizing inter-joint coordination and kinetic-chain force distribution—mechanisms relevant to movement efficiency and injury-risk management in multidirectional sports such as badminton [51]. From a risk–benefit standpoint, FCMT therefore appears theoretically well-suited for pre-competition programming. Additionally, the progressive load reduction in FCMT helps prevent fatigue accumulation from exceeding the enhancement effects of PAPE [52], thereby preserving neuromuscular recruitment capacity. This interpretation aligns with our fatigue monitoring, which showed lower post-session fatigue in FCMT compared with CT. With respect to high-load activation, FCMT's use of

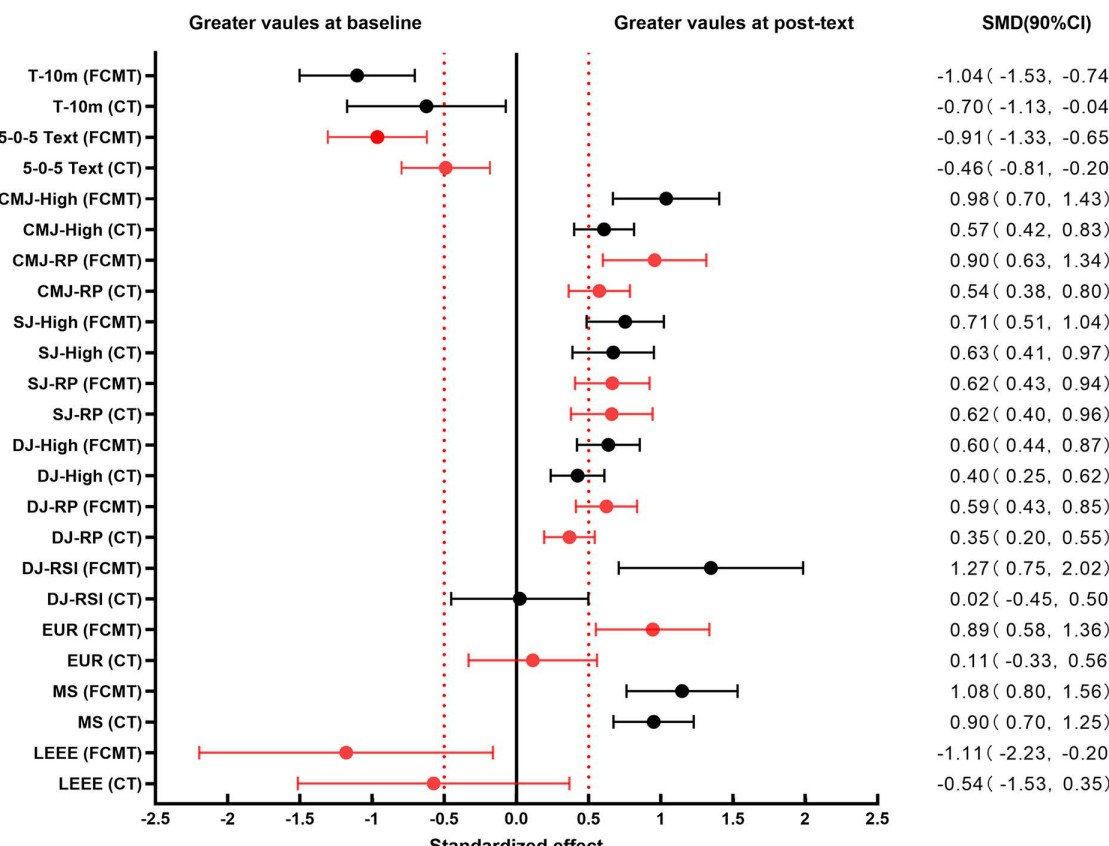

**Fig 4. Forest plot of standardized results for each test indicator before and after the experiment for both groups (90% confidence interval).**

isometric contractions is consistent with evidence that isometrics elicit robust PAPE responses [26–29], and may enhance the rate of force development (RFD)—a determinant associated with lower-limb explosive strength, reactive strength, and eccentric utilization rate [53–55]. Critically, we observed that the FCMT group demonstrated significantly greater improvements in these specific performance metrics compared to the CT group. Previous studies have demonstrated that isometric preactivation enhances subsequent peak force and explosive power output [56], providing additional supporting evidence despite being indirect. Finally, the present explosive-strength gains are congruent with reports of FCMT-related improvements in lower-body power among adult athletes across multiple sports [17–22]. These mechanistic interpretations are hypothesis-consistent rather than causal; studies incorporating neuromuscular measures (e.g., EMG timing, RFD profiling) are warranted to directly test these pathways.

In addition, differences in PAPE activation methods result in differences in the level of stimulation of the neuromuscular system within the entire training unit may be a plausible reason for the differences in SSC utilization gains between the two groups. It has been shown that lower extremity explosive strength gains are closely related to the utilization of elastic potential energy in the muscle and the activation of the muscle stretch reflex [49], and that the neuromuscular response varies with the level of fatigue [57]. Studies by C. Nichol et al. [58] have demonstrated that high-intensity anaerobic glycolytic exercise immediately attenuates the stretch reflex as a result of metabolic changes, thus decreasing sensitivity of the neural reflexes. In the high-load activation phase, CT employed the back squat exercise, while FCMT utilized the isometric back squat. The perceived exertion associated with the dynamic back squat in CT was reportedly greater than that produced by the isometric activation method in FCMT [59]. Relevant studies indicate that the onset of fatigue disrupts the functioning of the autonomic nervous system [60]. However, due to the accumulation of metabolites produced by fatigue caused an increase in the excitability of the Schwann cells, which in turn inhibited the vertebral alpha motoneurons from emitting nerve impulses, thereby reducing subsequent skeletal muscle power output [61]. In addition, the stabilizing muscles are subjected to greater moments of force during weighted squats with changes in joint angles (e.g., erector spinae), whereas the forces on the stabilizing muscles during isometric squats are constant and relatively small during exercise. Additionally, in the timing of training components, the FCMT maximizes the effects of PAP and PAPE by controlling decreasing loads to reduce fatigue buildup between cycles and achieve optimal exercise performance. Compared to CT, FCMT was likely more effective in activating the neuromuscular system and reducing fatigue buildup during training components such as the isometric back squat, back squat jump, and band-assisted jump. Coincidentally, the perceived exertion results of this study showed that the CT group consistently had higher levels of perceived exertion than the FCMT group, a finding that also indirectly confirms the above observations.

The results of the present study highlight the benefits of the FCMT in enhancing agility, and while this advantage may be attributed to enhanced explosive power or improved stretch-shortening cycle (SSC) capacity, another compelling reason may be due to the increased neuromuscular efficiency and coordination of the FCMT. Previous research has demonstrated that gains in athletes' agility capabilities may be attributed to enhanced neuromuscular efficiency and more effective utilization of the stretch-shortening cycle [62]. This mechanism is consistent with that proposed in this study. In addition, prior research has shown that there is a synergy between the components of the FCMT that seamlessly transmits force to activities with similar biomechanical requirements, especially those that require the generation of powerful thrusts from the hips and thighs [20]. Additionally, the advantages of FCMT for neuromuscular efficiency and coordination enhancement provide a more economical energy environment for explosive endurance gains, which is consistent with Cal Dietz et al. who suggest that FCMT can maintain greater power output over a longer period than CT, and provides a greater stimulatory effect to the body, with long-term adaptations leading to gains in speed endurance more significant [11].

This study utilized an uncommon and innovative assessment method to evaluate changes in subjects' explosive endurance. Subjects were assessed for the velocity dispersion, i.e., the standard deviation of the velocity of the 15-repetition weighted squat, at a weight of 30% 1RM (purpose: explosive power production is more favorable at this load [63]). This

innovative assessment method was chosen considering that the weighted squat is consistent in its movement pattern with the movements of the two training components. The results of the present study demonstrated the advantage of the FCMT in terms of an increase in explosive endurance, a result that is consistent with the results of two previous studies. Noufal K V et al. [20] found that the FCMT significantly improved anaerobic strength capacity in field hockey players by performing the FCMT on 15 field hockey players over 12 weeks. Chang et al. [64] performed a randomized controlled trial on 24 college students, and explored the effects of 8 weeks of high-intensity strength training (similar to the training structure of FCMT) and resistance training on athletic performance, and one of the results showed that this high-intensity strength training produced better improvements in mean anaerobic power compared to traditional resistance training, which is consistent with the findings of the present study. It is also worth considering that the adaptation period for CT was approximately 3 weeks compared to 1 month for FCMT, so athletes may have adapted to CT earlier than to FCMT, and the training effect of CT seems to be more dependent on the accumulation of load intensity. However, premature adaptation to CT in the CT group during the 8-week training intervention may also have reduced the degree of stimulation in subsequent training, leading to differences in explosive endurance gains between the two groups.

While the FCMT demonstrated clear superiority in metrics involving the explosive power and its endurance, it is noteworthy that both training protocols were equally effective in enhancing maximum strength (1RM squat), speed-strength qualities as measured by the Squat Jump (SJ), and sprinting acceleration as measured by the 10-meter sprint (T-10m). The absence of statistically significant between-group differences in these outcomes (p > 0.05 for 1RM, SJ, and T-10m metrics, Table 6) indicates that the foundational capacities for generating high levels of concentric force and initial acceleration were developed to a similar extent by both CT and FCMT. This finding underscores that the high-load component inherent to both protocols, the back squat in CT and the isometric back squat in FCMT, provided a sufficient stimulus for eliciting comparable neuromuscular and structural adaptations related to maximal force production. Consequently, for coaches whose primary aim in a training cycle is to build a solid strength base, improve concentric power, or enhance short-distance sprint performance, both CT and FCMT appear to be viable and effective strategies. Taken together, FCMT's distinct advantage does not lie in superior development of primary concentric strength or linear sprint velocity, but rather in enhancing EUR and reactive strength, thereby optimizing the transfer of strength to subsequent reactive, explosive and its endurance-oriented tasks. Notably, all participants were elite adolescents with ≥3 years of systematic resistance-training experience; such training age likely conferred sufficient neuromuscular adaptation and recovery capacity to manage fatigue within FCMT's sequential four-exercise structure, reducing the risk that potentiation effects would be obscured by cumulative fatigue. By contrast, the more discrete paired structure of CT may be preferable for novices or athletes with limited training background.

Improvements in either agility, reactive strength, or SSC utilization are effective in enhancing badminton attacking efficiency and reducing joint injuries associated with multi-angle direction changes [5,65]. In addition, the improvement in explosive endurance was effective in enhancing the duration of athletic performance [13,66]. This study has several advantages, as the FCMT group showed significant improvement in lower limb explosive strength, reactive strength, agility, SSC utilization, and explosive endurance compared to the CT group, which suggests that FCMT is effective in enhancing the athletic performance of junior badminton players. The results of this study are consistent with previous research and extended to explore comparisons of explosive endurance, adding credibility and providing coaches with practical suggestions for incorporating FCMT into their training regimens. However, there are some limitations to this study: A) Due to the scarcity of the elite junior badminton athlete population, this study suffers from a small sample size and was conducted only among male athletes. B) The load intensity was not reset in the middle of the experiment considering the accuracy of fatigue monitoring and the purpose of exploring the training acclimatization period, and subsequent studies could further explore the actual effect after resetting the load in the middle of the experiment for the training acclimatization period. C) Considering the consistency between the training actions and the testing actions, an uncommon and innovative assessment method was used to assess explosive endurance changes, and although it has been cited in previous studies, the

effectiveness of its assessment and its practical application value still need to be further verified through more research. D) The exercises within the interventions were different. As a result, it is unclear whether the differences observed are due to the different training structures, or simply the different exercises. E) Sleep patterns of participants were not monitored throughout the study. Given that sleep duration and quality significantly influence cognitive function, reaction time, and the ability to produce maximal efforts during both training and testing, especially in tasks requiring explosive power and sustained attention, the lack of sleep control represents a potential confounding variable [67]. Future research should incorporate sleep monitoring to better account for its role in training adaptation and performance outcomes. Despite these limitations, this study effectively demonstrated the value of the FCMT in improving athletic performance.

Based on the results of this study, it is recommended that French contrast method training (FCMT) be incorporated into a training program for adolescent male badminton players. An eight-week intervention program can be designed to adjust the loading intensity in four-week cycles, focusing on the improvement of explosive power, agility, and explosive endurance. Despite the limitations of the study, personalized training programs need to combine FCMT with badminton-specific training and monitoring to guarantee the improvement of athletic performance to meet the athletes' stage-by-stage fitness needs, as well as to establish athlete profiles with the test data for personalized teaching. Coaches should pay more attention to the alternation of load intensity when arranging complex training, and pay more attention to the timing of training when arranging the French contrast method training. Combining the advantages and limitations of the two training methods, it is recommended that coaches use CT to develop the basic explosive power in the early stage of developing the explosive power quality of young athletes, and arrange FCMT in the strength training in the pre-competition training period to improve the athletes' performance and competitive level. In addition, it is recommended to choose isometric back squat as a means of high-load activation for FCMT. In summary, this practical approach can lead to a sustained improvement in the competitive performance of badminton players, which is in line with the modern pace of the game.

## Conclusion

The present study provides compelling evidence that the French Contrast Method Training (FCMT) is significantly more effective than Complex Training (CT) in enhancing lower limb explosive strength, reactive strength, agility, stretch-shortening cycle (SSC) utilization, and explosive endurance in adolescent male badminton athletes within a short-term training period. While both training modalities improved maximum strength and speed-related performance, FCMT produced superior outcomes across a broader range of neuromuscular adaptations. These advantages may be attributed to the FCMT's organizational structure and maximizing PAPE activation methods. Additionally, fatigue monitoring revealed that although FCMT had a longer adaptation period than CT, it led to a greater overall performance improvement with a lower cumulative fatigue load. These findings support the integration of FCMT into pre-competition training phases, particularly for youth athletes requiring rapid and sustained explosive efforts, such as those in badminton. Coaches are encouraged to apply FCMT in a periodized manner alongside sport-specific drills to achieve individualized, performance-oriented outcomes.

## Acknowledgments

The authors declare that the research was conducted in the absence of any commercial or financial relationships that no conflicts of interest.

## Author contributions

**Conceptualization:** Ruiyin Huang, Yuhua Gao, Zhan Gao.

**Data curation:** Ke Yang, Yong Mo.

**Investigation:** Yongren Lu.

**Methodology:** Ruiyin Huang, Yuhua Gao, Zhan Gao.

**Project administration:** Yuhua Gao, Yong Mo.

**Software:** Ruiyin Huang.

**Supervision:** Ke Yang, Yong Mo.

**Validation:** Yongren Lu.

**Visualization:** Ruiyin Huang, Yongren Lu.

**Writing – original draft:** Ruiyin Huang.

**Writing – review & editing:** Yuhua Gao, Ke Yang, Yong Mo, Zhan Gao.

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
