## [Decision Letter · Decision Letter 0]

6 Oct 2025

Dear Dr. Gao,

Thank you for submitting your manuscript to PLOS ONE. After careful consideration, we feel that it has merit but does not fully meet PLOS ONE’s publication criteria as it currently stands. Therefore, we invite you to submit a revised version of the manuscript that addresses the points raised during the review process.

We look forward to receiving your revised manuscript.

Kind regards,

Javier Abián-Vicén, Ph.D.

Academic Editor

PLOS ONE

Journal Requirements:

When submitting your revision, we need you to address these additional requirements

Additional Editor Comments:

Dear Authors,

We appreciate the considerable effort you have made in revising your manuscript and acknowledge that the topic is of clear interest to our readership. However, after carefully considering the reviewers’ feedback, we have concluded that the paper requires a major revision before it can be considered for publication. Both reviewers agree that, while the study shows potential, there are substantial methodological issues and limitations in the experimental design and interpretation of the findings that must be thoroughly addressed. In particular, it is essential to reframe the discussion and conclusions, explicitly acknowledging the implications of using a fixed training load and how this affects the external validity of your results. We ask that you carefully respond to the reviewers’ comments, incorporate the suggested references, and revise both the methodological framework and the overall argumentation to strengthen the scientific rigor of the manuscript and avoid interpretations that could mislead coaches and practitioners.

Reviewer's Responses to Questions

**Comments to the Author**

1. Is the manuscript technically sound, and do the data support the conclusions?

Reviewer #1: No

Reviewer #2: Partly

2. Has the statistical analysis been performed appropriately and rigorously?

Reviewer #1: Yes

Reviewer #2: Yes

3. Have the authors made all data underlying the findings in their manuscript fully available?

Reviewer #1: No

Reviewer #2: Yes

4. Is the manuscript presented in an intelligible fashion and written in standard English?

Reviewer #1: Yes

Reviewer #2: Yes

Reviewer #1: Broad comments:

There is a notable limitation with this paper whereby the exercises between the two interventions are different. As a result, it is unclear whether the observed changes are due to the structure of the session (i.e., CT vs FCT) or simply due to the differences in exercises. This needs to be acknowledged more clearly. Moreover, the extent to which the exercises within the conditions are different (i.e., load, tempo, etc) is not clear. It is also likely that this has influenced the outcomes.

The introduction and discussion also read like advertisements for French contrast training. These need to be revised to be more balanced and provide a more accurate representation of the literature.

There are some deviations form your trial registration that need to be rectified and/or acknowledged in the methods.

Introduction:

Line 35 – this first sentence also needs some commentary around the importance of power. It is really the need to improve both simultaneously that has created the interest in contrast training methods.

Line 37 – building on the prior point, it would be good to expand on this. The reasons it is becoming popular is not because it is the best way t improve strength. It is because it is suggested to offer an effective way to improve strength and power simultaneously.

Line 37 – it would also be good to introduce complex training more broadly before describing French contrast and complex contrast (which is what you are examining). The review paper by Comier is a great place to start (DOI: 10.1007/s40279-022-01715-x)

Line 38 – you need to explain what PAP is. It would also be more appropriate to describe it as post-activation performance enhancement (PAPE) as that is what you are talking about.

Line 41 – need to temper your language here. While there is evidence to suggest that the PAPE response does improve performance in the short term, there is limited to suggest that these modes of training offer any additional advantages in the long term. It is “theorised” to enhance performance more than traditional training methods.

Line 51 to 56 – this reads like an advertisement. All of this is theoretical. Suggest removing and/or tempering your language and accurately representing the research on FCT, and removing buzzwords.

Line 68 – remove “centripetal and centrifugal”

Line 73 – remove “(i.e., complex training)” as neither of these studies looked at complex training.

Line 74 to 75 – this citation is not appropriate for this statement, and it is only partially accurate. A Delphi study of elite strength and condition coaches by Luders et al., 2024 (https://doi.org/10.1177/17479541241272256) suggested that it is used because it is time efficient. A follow up intervention found complex training to be equally as effective at improving athletic performance as normal strength and power training, albeit took less time to complete (https://doi.org/10.1519/JSC.0000000000004888). A more balanced discussion of the potential benefits of these types of training methods supported by the above studies would be better here.

Line 78 – This aim/objective is different to your trial registration, where you explicitly state that you expect FCMT to be better. Please revise in line with the registration.

Methods:

Line 88 – can you please detail how they were “adapted” to both interventions?

Line 89 – this method of randomisation is different to that described in the pre-registration. Please revise or add a section title “deviations from pre-planned protocol” to highlight these deviations.

Ine 135 – you need to justify your exercise selection here. Why were they different? Also need to provide some explanation on how they were performed (tempo etc) so that any differences are really clear.

General comment – it also looks like a few tests you looked at were not included in your trial registration. These should also be notes when describing other deviations.

General comment – in your registration you state the data will be made publicly available. It should be noted here with the link to the repository provided.

General comment – statistical analysis. There is a huge number of analyses conducted here. You should do a post-hoc correction for multiple comparisons, as some of the between group differences may just be through chance due to your small sample and large number of comparisons.

Results:

General comment – would be good to report mean and SD for session adherence here somewhere as well.

Line 239 – remove the word autonomic from this section. You are simply assessing perceived effort. Moreover, none of this was compared statistically, so this cannot really be stated. The figure looks as if the changes/differences are small, and unlikely meaningful. Suggest removing commentary around the differences.

Discussion:

General comment – the discussion focuses on FCMTs “superiority” by a bunch of unsubstantiated mechanisms (for example, “neuromechanical advantages across the force–velocity spectrum,”) but no direct neural or mechanical data was collected. This is just speculation. As such, the language throughout the discussion needs to reflect that these are potential explanations rather than conclusions.

General comment – the outcomes whereby there were no between-group differences is downplayed. Both methods seem equally effective for maximal strength and concentric, which should be mentioned in more detail.

Line 286 – not sure if “plate” is the correct word here. Maybe stage or section?

Line 287 to 291 – so much of this is non-scientific and just buzzwords. I would suggest highlighting some of the differences between the interventions as a way to discuss some potential difference sin outcomes, rather than just making statements about the hypothetical benefits of FCT.

Line 357 – another limitation that need to be added: A) The exercises within the interventions were different. As a result, it is unclear whether the differences observed are due to the different training structures, or simply the different exercises.

Conclusion:

Line 385 – suggest replacing “can” with “may” as you don’t know the reason.

Would also be good to add something like “Future studies should match exercises and volume between conditions to isolate the effects of training structure.”

Table 1 – the exercises in this table are different to that in the methods. Please revise for consistency and clarify.

Figure 3 – add standard deviations to each time point

Reviewer #2: General Comments

This manuscript presents a single-blind, randomized controlled trial comparing the effects of an 8-week French Contrast Method Training (FCMT) versus Complex Training (CT) on measures of explosive power and endurance in elite adolescent male badminton players. The authors conclude that FCMT is superior to CT for improving reactive strength, agility, SSC efficiency, and explosive endurance. While the study addresses a relevant and practical question in sports science and the manuscript has been improved from a previous version, several major and minor weaknesses remain that require attention before it can be considered for publication.

Major Weaknesses

Fixed Training Load Protocol: The most significant methodological weakness is the use of a fixed load for the entire 8-week intervention period . For elite adolescent athletes, the principle of progressive overload is fundamental for inducing continued adaptation. An 8-week training block without any load progression is highly likely to lead to a training plateau. The study's own RPE data (Figure 3) suggests the CT group adapted much earlier (around week 3) than the FCMT group . This lack of progressive overload may have acted as a confounding variable, artificially limiting the potential gains of the CT group and biasing the results in favor of the more novel and complex FCMT protocol, which likely required a longer adaptation period. This profoundly impacts the interpretation and external validity of the findings.

Validity of the Lower Extremity Explosive Endurance (LEEE) Test: The primary test for explosive endurance is described as "innovative" but lacks sufficient validation. It involves measuring the velocity dispersion of 15 half-squats at 30% 1RM . The rationale for this specific protocol is not well-supported by robust, peer-reviewed literature; the main citation appears to be a thesis . Furthermore, the reliability of this novel test was not assessed , which is a critical omission for an outcome measure used to draw significant conclusions.

Minor Weaknesses

Small Sample Size: While the authors have now included a power analysis and provided justification by citing similar studies , the sample size of n=10 per group remains small, which limits the statistical power and the generalizability of the results.

Participant Screening Description: The manuscript states an inclusion criterion of "no impairment in squatting movement patterns (over-the-shoulder squat score of at least 2)" . The authors’ rebuttal letter to previous reviewers clarifies that only the overhead squat component of the FMS was used and all participants scored a 3. This level of detail is necessary in the manuscript itself for clarity and transparency regarding the functional homogeneity of the sample.

Data Visualization: In Figure 3, the decision to omit error bars (e.g., SD) in favor of a data table below the chart reduces the visual effectiveness of the graph. Error bars are standard practice and essential for visually interpreting the variability and overlap in perceived exertion between the groups over time.

Specific Comments

Abstract

Page 8, Lines 124-126: The statement "...FCMT showed a delayed but more substantial performance enhancement after 4 weeks, with lower cumulative RPE scores" could be misinterpreted. The RPE scores for FCMT were consistently lower throughout, not just cumulatively . The key finding is the differing adaptation timelines, which should be stated more clearly.

Introduction

Page 11, Lines 185-186: The introduction could be strengthened by providing a more detailed biomechanical rationale for why explosive power is critical in badminton. Citing recent work on muscle synergy and neuromuscular coordination during key movements like the overhead smash would provide a stronger foundation for investigating advanced training methods that target these qualities. It is recommended to include the following reference:

[Tajik R, Dhahbi W, Fadaei H, Mimar R: Muscle Synergy Analysis During Badminton Forehand Overhead Smash: Integrating Electromyography and Musculoskeletal Modeling. Frontiers in Sports and Active Living 2025, 7:1596670.]

Methods

Page 14, Lines 258-259: As stated in the major weaknesses, the sentence "Both groups were trained with a fixed load... during the 8-week intervention period" details a significant methodological flaw . The rationale to "observe the trend of subjects' adaptation" does not justify violating the fundamental training principle of progressive overload .

Page 20, Lines 456-458: In the Maximum Strength Test subsection, the direct measurement protocol for 1RM is described . To bolster the methodological rigor, it is recommended to cite recent literature that validates or discusses protocols for direct 1RM testing in athletic populations. The following reference would be appropriate:

[Dhahbi W, Padulo J, Russo L, Racil G, Ltifi M-A, Picerno P, Iuliano E, Migliaccio GM: 4-6 Repetition Maximum (RM) and 1-RM Prediction in Free-Weight Bench Press and Smith Machine Squat Based on Body Mass in Male Athletes. Journal of strength and conditioning research 2024.]

Page 22, Lines 502-503: The explicit statement that the Maximum Strength and LEEE tests were not included in reliability assessment is a notable weakness for the LEEE test . While understandable for a 1RM test, the lack of reliability data for a novel primary outcome measure is problematic.

Discussion

Page 26, Lines 603-605: When discussing the superior agility gains in the FCMT group, the manuscript would benefit from comparing its findings with other recent studies that have investigated combined plyometric and change-of-direction training in elite youth athletes. This would provide a richer context for interpreting the results. The authors should consider discussing their findings in light of this study:

[Cherni Y, Mzita I, Oranchuk DJ, Dhahbi W, Hammami M, Ceylan HI, Stefanica V, Chelly MS: Effects of loaded vs unloaded plyometric training combined with change-of-direction sprints on neuromuscular performance in elite U-18 female basketball players: a randomized controlled study. Sport Sciences for Health 2025:1-13.]

Page 29, Lines 686-689: The discussion proposes that FCMT's benefits stem from its broad stimulation across the force-velocity curve . This could be expanded by framing the holistic nature of FCMT within a modern paradigm like the 'joint-by-joint' training approach, which emphasizes inter-joint coordination and could also have implications for injury prevention in a multi-directional sport like badminton. Incorporating this perspective is recommended:

[Dhahbi W, Materne O, Chamari K: Rethinking knee injury prevention strategies: joint-by-joint training approach paradigm versus traditional focused knee strengthening. Biology of Sport 2025, 42(4):59-65.]

Page 31, Lines 738-744: The limitations section should acknowledge other uncontrolled variables that can significantly influence training adaptation and performance. For instance, the manuscript does not mention monitoring athlete sleep. Given that factors like sleep duration can impact physical and cognitive performance, this should be noted as a potential confounder and an area for future research. It is suggested to add this point and cite the following work:

[Bouzouraa E, Dhahbi W, Ferchichi A, Geantă VA, Kunszabo MI, Chtourou H, Souissi N: Single-Night Sleep Extension Enhances Morning Physical and Cognitive Performance Across Time of Day in Physically Active University Students: A Randomized Crossover Study. Life 2025, 15(8):1178.]

**Do you want your identity to be public for this peer review?** For information about this choice, including consent withdrawal, please see our Privacy Policy

Reviewer #1: No

Reviewer #2: **Yes: ** Wissem Dhahbi

---

## [Author Response · Author response to Decision Letter 1]

21 Oct 2025

Dear Editors of PLOS ONE,

We sincerely thank the Academic Editor and both reviewers for their thoughtful critiques and the opportunity to submit a revised manuscript. In response,we implemented revisions to improve theoretical framing, methodological transparency, and interpretive balance. Key changes include: (1) strengthening the theoretical framework in the Introduction and Discussion; (2) explicitly justifying methodological choices (e.g., fixed loads; differing exercises) and acknowledging their implications; (3) integrating all suggested citations; (4) tempering claims to align strictly with the data and study limitations; and (5) updating data and figure compliance. The dataset has been re-uploaded to Figshare (DOI: 10.6084/m9.figshare.30349558), and figures were checked with PACE.

Below is a detailed point-by-point response.

Response to Reviewer #1:

We sincerely thank Reviewer #1 for their insightful and constructive comments, which have been instrumental in strengthening our manuscript. First, concerning the use of different exercises between the FCMT and CT groups, we acknowledge this as an inherent aspect of comparing these distinct training methodologies. The French Contrast Method Training is structured around a specific four-exercise sequence designed to systematically exploit post-activation performance enhancement (PAPE), which necessarily involves different exercises and loading patterns than the paired-exercise structure of traditional Complex Training. We recognize that this difference introduces a potential confounding variable. Accordingly, we have clarified the rationale for our exercise selection in the Methods and explicitly acknowledged this limitation in the Discussion, recommending that future studies match exercises to isolate the effect of training structure. Second, we agree that the tone in the original Introduction and Discussion could be perceived as overly promotional. We have thoroughly revised these sections to adopt a more balanced and scientific perspective. This includes tempering claims about FCMT, citing relevant literature such as the review by Cormier et al. to provide context, and using more cautious language (e.g., "may," "suggest") when interpreting results. We have also emphasized outcomes where both training methods were equally effective to ensure a balanced presentation. Our point-by-point responses below detail all specific changes made in the manuscript.

Comment 1:

Introduction:

(1)Line 35 – this first sentence also needs some commentary around the importance of power. It is really the need to improve both simultaneously that has created the interest in contrast training methods.

(2)Line 37 – building on the prior point, it would be good to expand on this. The reasons it is becoming popular is not because it is the best way t improve strength. It is because it is suggested to offer an effective way to improve strength and power simultaneously.

(3)Line 37 – it would also be good to introduce complex training more broadly before describing French contrast and complex contrast (which is what you are examining). The review paper by Comier is a great place to start (DOI: 10.1007/s40279-022-01715-x).

(4)Line 38 – you need to explain what PAP is. It would also be more appropriate to describe it as post-activation performance enhancement (PAPE) as that is what you are talking about.

(5)Line 41 – need to temper your language here. While there is evidence to suggest that the PAPE response does improve performance in the short term, there is limited to suggest that these modes of training offer any additional advantages in the long term. It is “theorised” to enhance performance more than traditional training methods.

(6)Line 51 to 56 – this reads like an advertisement. All of this is theoretical. Suggest removing and/or tempering your language and accurately representing the research on FCT, and removing buzzwords.

(7)Line 68 – remove “centripetal and centrifugal”

(8)Line 73 – remove “(i.e., complex training)” as neither of these studies looked at complex training.

(9)Line 74 to 75 – this citation is not appropriate for this statement, and it is only partially accurate. A Delphi study of elite strength and condition coaches by Luders et al., 2024 (https://doi.org/10.1177/17479541241272256) suggested that it is used because it is time efficient. A follow up intervention found complex training to be equally as effective at improving athletic performance as normal strength and power training, albeit took less time to complete (https://doi.org/10.1519/JSC.0000000000004888). A more balanced discussion of the potential benefits of these types of training methods supported by the above studies would be better here.

(10)Line 78 – This aim/objective is different to your trial registration, where you explicitly state that you expect FCMT to be better. Please revise in line with the registration.

Response:

(1)We have restructured the opening section of the introduction based on the reviewers' suggestions. The revised content first explicitly defines badminton as a multidirectional explosive sport, noting that outstanding performance depends not only on strength and power but also on highly efficient neuromuscular coordination. Building on this foundation, we further emphasize that the distinct physiological demands of strength and power during training adaptation make their simultaneous development challenging. This naturally leads to the introduction of the Post-Activated Performance Enhancement (PAPE) mechanism and its derivative methods, such as Complex Training (CT) and the French Contrast Method of Training (FCMT). (Lines 37-53)

(2)Synchronized with the modifications to (1).

(3)We have incorporated the reviewers' suggestions by providing a broader introduction to complex training before describing the two training methods. After carefully reviewing the study by Comier et al., we found it effectively explains the concept and principles of wind complex training, and have therefore cited it. (Lines 45-47)

(4)Indeed, describing it as PAPE would be more appropriate, as most of our test metrics are used for performance evaluation. We have explained PAPE in the text. (Lines 43-45)

(5)Indeed, in our original manuscript, we overemphasized the description of FCMT and became overly “theoretical” when discussing long-term effects, lacking sufficient evidence. Therefore, we have revised our wording. (Lines 50-52)

(6)We have removed the previous version and made adjustments to the description of FCMT. (Lines 50-52)

(7)We have removed it.

(8)We have removed it.

(9)We sincerely thank the reviewers for providing us with two constructive references. We have reviewed our previous citations and found that we did not sufficiently substantiate the claim that “its effectiveness has been widely recognized by coaches.” Therefore, we have incorporated the two references suggested by the reviewers, which provide robust evidence for the time efficiency and performance benefits of CT from both the coaches' and athletes' perspectives. (Lines 64-66)

(10)We have adjusted the description of the objectives based on the trial registration. (Lines 69-71)

Comment 2:

Methods:

(1)Line 88 – can you please detail how they were “adapted” to both interventions?

(2)Line 89 – this method of randomisation is different to that described in the pre-registration. Please revise or add a section title “deviations from pre-planned protocol” to highlight these deviations.

(3)Ine 135 – you need to justify your exercise selection here. Why were they different? Also need to provide some explanation on how they were performed (tempo etc) so that any differences are really clear.

(4)General comment – it also looks like a few tests you looked at were not included in your trial registration. These should also be notes when describing other deviations.

(5)General comment – in your registration you state the data will be made publicly available. It should be noted here with the link to the repository provided.

(6)General comment – statistical analysis. There is a huge number of analyses conducted here. You should do a post-hoc correction for multiple comparisons, as some of the between group differences may just be through chance due to your small sample and large number of comparisons.

Response:

(1)We acknowledge that the term “adapted” was used incorrectly; our intent was to convey ‘familiar’ (lines 79-81). The wording in this section has been revised, and the training details are described in the “Training program” section at lines 143-144.

(2)We have modified the pre-registered randomization method to ensure consistency. (Lines 82-88)

(3)We restructured the description of the “Training program” section to clarify the rationale behind exercise selection. We explained why different exercises were chosen by discussing the fundamental differences between the two training approaches (organizational structure and maximizing PAPE) (lines 123-132). Additionally, we supplemented details on exercise tempo and set execution (lines 139-144).

(4)We have removed two metrics from the previously submitted manuscript: the standing long jump and the Pre-stretch augmentation percentage (PSAP). This addresses the previous reviewer's suggestion that our test metrics overlapped. Between the standing long jump and the vertical jump, we selected the vertical jump due to its superior ability to reflect lower-limb dynamics. Additionally, between the Eccentric Utilization Ratio (EUR) and PSAP, we selected EUR because it is more oriented toward assessment, whereas PSAP is more suited for diagnosing personalized training protocols.

(5)Our data is provided in the “Date Availability Statements” section. “The data that support the findings of this study are available in Figshare at http://doi.org/10.6084/m9.figshare.30349558.”

(6)Given the small sample size, we adopted the reviewers' suggestion to perform post-hoc corrections for multiple comparisons. Tukey's method was used for post-hoc tests when data followed a normal distribution, while the Dwass-Steel-Critchlow-Fligner method was applied when data did not follow a normal distribution. The corrected results did not negate the original conclusions. (lines 237-240)

Comment 3:

Results:

(1)General comment – would be good to report mean and SD for session adherence here somewhere as well.

(2)Line 239 – remove the word autonomic from this section. You are simply assessing perceived effort. Moreover, none of this was compared statistically, so this cannot really be stated. The figure looks as if the changes/differences are small, and unlikely meaningful. Suggest removing commentary around the differences.?

Response:

(1)Regarding session adherence, all participants in the trial completed all training and testing sessions without any absences or dropouts. This point is also mentioned in lines 247-249. (No subjects reported adverse effects due to injury or early withdrawal from the study over the course of the 8-week study, resulting in a total of 20 subjects (10 in the FCMT group and 10 in the CT group) included in the metrics data analyzed statistically.)

(2)We have removed the term “autonomic.” However, we believe analysis of fatigue monitoring is necessary. Although we did not perform statistical comparisons here, the plotted “average change curve of RPE scale indices after 16 training sessions for both groups” (Figure 3) clearly shows that the difference in perceived exertion post-training between the two groups increased after the 9th session. We have also re-added error bars to Figure 3, which allows for more intuitive observation of this point. (lines 254-261)

Comment 4:

Discussion:

(1)General comment – the discussion focuses on FCMTs “superiority” by a bunch of unsubstantiated mechanisms (for example, “neuromechanical advantages across the force–velocity spectrum,”) but no direct neural or mechanical data was collected. This is just speculation. As such, the language throughout the discussion needs to reflect that these are potential explanations rather than conclusions.

(2)General comment – the outcomes whereby there were no between-group differences is downplayed. Both methods seem equally effective for maximal strength and concentric, which should be mentioned in more detail.

(3)Line 286 – not sure if “plate” is the correct word here. Maybe stage or section?

(4)Line 287 to 291 – so much of this is non-scientific and just buzzwords. I would suggest highlighting some of the differences between the interventions as a way to discuss some potential difference sin outcomes, rather than just making statements about the hypothetical benefits of FCT.

(5)Line 357 – another limitation that need to be added: A) The exercises within the interventions were different. As a result, it is unclear whether the differences observed are due to the different training structures, or simply the different exercises.

Response:

(1)We agree with the reviewer's perspective. For a series of unproven mechanisms, we have adjusted the wording of our discussion by using terms such as “potentially” and “seem to” to reflect that these represent potential explanations rather than definitive conclusions.

(2)We greatly appreciate the reviewer's insightful suggestion. This was indeed an oversight on our part. We have added a paragraph describing this in lines 387–398.

(3)Our original intent was to include a section. However, considering that this part of the description was overly subjective and lacked supporting evidence, we deleted it and rewrote the part. (lines 298-324)

(4)As stated in (3), we rewrote this section because we considered it overly subjective and lacking direct evidence. We first explicitly clarified that the efficacy difference between FCMT and CT stems from their distinct training structures and activation methods, establishing a comparative foundation by citing methodological details. Subsequently, we conducted a multidimensional mechanism analysis: FCMT's load progression sequence provides more comprehensive stimulation of the force-velocity curve (referencing Dietz C et al.'s “Triphasic training: A systematic approach to elite speed and explosive strength performance”), aligning with modern joint coordination training principles (incorporating Reviewer 2's suggestions) to enhance kinetic chain efficiency and movement economy; simultaneously, the external load reduction strategy delays neuromuscular fatigue, supported by subjective fatigue data. Subsequently, by linking the training method to outcome measures such as reaction force and SSC efficiency through the relationship between the isometric contraction characteristics of static squats and the enhancement of rate of force development (RFD), and by citing literature to establish the theoretical connection between RFD and lower-body explosive qualities. Finally, multiple athlete studies are referenced to substantiate the argument, completing the closed-loop reasoning from methodological differences → physiological mechanisms → empirical results → theoretical elevation. (lines 298-324)

(5)This is necessary, so we have added “D)” to the restrictions section. (lines 380-381)

Comment 5:

Conclusion:

(1)Line 385 – suggest replacing “can” with “may” as you don’t know the reason.

(2)Would also be good to add something like “Future studies should match exercises and volume between conditions to isolate the effects of training structure.”

(3)Table 1 – the exercises in this table are different to that in the methods. Please revise for consistency and clarify.

(4)Figure 3 – add standard deviations to each time point.

Response:

(1)Based on the feedback, we have made adjustments to the wording.

(2)We have added this suggestion in lines 397-399.

(3)We reviewed the statements in Table 1 and the Methods section and made revisions to the wording in the Methods section. (lines 133-138)

(4)We have added error bars to Figure 3.

Response to Reviewer #2:

We sincerely thank Reviewer #2 for their thorough and insightful evaluation and for acknowledging the improvements made to the manuscript. Their comments on methodological weaknesses were particularly valuable in enhancing the scientific rigor and clarity of our work. Regarding

---

## [Decision Letter · Decision Letter 1]

17 Nov 2025

Dear Dr. Gao,

Thank you for submitting your manuscript to PLOS ONE. After careful consideration, we feel that it has merit but does not fully meet PLOS ONE’s publication criteria as it currently stands. Therefore, we invite you to submit a revised version of the manuscript that addresses the points raised during the review process.

We look forward to receiving your revised manuscript.

Kind regards,

Javier Abián-Vicén, Ph.D.

Academic Editor

PLOS ONE

Journal Requirements:

Reviewers' comments:

Reviewer's Responses to Questions

**Comments to the Author**

Reviewer #1: All comments have been addressed

Reviewer #2: (No Response)

2. Is the manuscript technically sound, and do the data support the conclusions?

Reviewer #1: Yes

Reviewer #2: (No Response)

3. Has the statistical analysis been performed appropriately and rigorously?

Reviewer #1: Yes

Reviewer #2: (No Response)

4. Have the authors made all data underlying the findings in their manuscript fully available?

Reviewer #1: No

Reviewer #2: (No Response)

5. Is the manuscript presented in an intelligible fashion and written in standard English?

Reviewer #1: Yes

Reviewer #2: (No Response)

Reviewer #1: I would like to thank the authors for taking the time to respond to my comments so comprehensively. I have a couple of additional minor comments.

Abstract:

Line 13 – suggest changing “Compared with the Complex Training methods that have significant and widespread effects on enhancing explosive power…” to “While evidence have indicated complex contrast training can enhance strength and power…”

Introduction:

Line 50 – suggest changing “self-weight” to “bodyweight jumps” and “reduced load” to “assisted jumps”

Line 60 – should be stretch-shortening cycle (not stretch-short)

Methods:

Line 117 – the sessions were separated by closer to 72 hours minimum?

Line 235 – typo of after the Tukey method (..)

Discussion:

Line 328 – I would suggest revising the final two sentences of this paragraph. You didn’t measure fatigue, you measured perceived exertion, which are quite different constructs.

Line 333 – typo here (,,)

Conclusion:

Line 406 – it would be good to note that it was better over a short time frame, as the study was only 8 weeks (rather than confidently saying it is simply better)

Data availability statement:

The link you have shared does not work. Please correct to provide raw data.

Table 1 – need to capitalise the words “isometric” and “back”

Table 3, 4, and 5 – suggest changing “MS” to “1RM Squat”

Reviewer #2: General Comments

The authors have comprehensively addressed the major concerns raised in the initial review. The theoretical framework in the Introduction and Discussion is much stronger, methodological justifications have been provided, and all suggested citations have been thoughtfully integrated. The manuscript is substantially more balanced and scientifically rigorous as a result.

Major Weaknesses (Now Acknowledged as Limitations): The primary methodological weaknesses of the study—namely the use of a fixed training load and the non-matched exercises between intervention groups —remain. However, the authors have now appropriately acknowledged these as significant limitations in the Discussion section. This is a satisfactory response for a completed trial, as these design elements cannot be changed post-hoc.

Minor Weaknesses (Remaining): The manuscript's quality is much improved, but it still requires minor copy-editing for clarity, consistency, and the correction of a few oversights from the revision process. The most significant remaining issue is a direct contradiction in the data availability statements (see specific comments).

Specific Comments

Introduction

Page 18, Line 348: The phrase "strong muscle contractions, as well as effective utilization of the stretch-shorten cycle" is clear. However, the track-changes document indicates the authors intended to remove "centripetal and centrifugal," as agreed upon in their response to Reviewer #1 (Point 7) , but this phrase was not fully removed and remains in the clean version. Please remove it as agreed.

Methods

Page 12, Line 221 (and multiple other locations): There is a significant contradiction regarding data availability.

The "Response to Reviewers" letter states the data is on Figshare (DOI: 10.6084/m9.figshare.30349558).

The "Date Availability Statements" section in the main manuscript (Page 42, Line 988) also lists this Figshare DOI.

However, the "Data Availability" section of the submission form (Page 12, Line 221) states: "All the data obtained from the experiments can be retrieved from the ScienceDB database (dataset number: 10.57760/sciencedb. 18250)".

Please resolve this contradiction and ensure the correct repository and DOI/accession number are stated consistently throughout all parts of the submission.

Discussion

Page 40, Line 956: The "Practical Applications" section recommends "isometric back squat". This is inconsistent with the term used in the Conclusion. Please ensure consistent terminology.

Conclusion

Page 16, Line 289 (from Abstract): There is a typographical error: "The findings suggest that The French Contrast..." This should be corrected to "The findings suggest that the French Contrast..."

Page 42, Line 972 / Page 101, Line 2366: The authors correctly adopted the suggestion to change "can" to "may" ("These advantages may be attributed..."). However, the conclusion (Page 40, Line 956) and "Practical Applications" (Page 101, Line 2362) use the term "isometric static deep-back squat." This terminology is not used in the Methods section or Table 1, which use "isometric back squat." Please standardize this terminology throughout the manuscript for clarity.

**Do you want your identity to be public for this peer review?** For information about this choice, including consent withdrawal, please see our Privacy Policy

Reviewer #1: No

Reviewer #2: **Yes: ** Wissem Dhahbi

---

## [Author Response · Author response to Decision Letter 2]

18 Nov 2025

Dear Editors of PLOS ONE,

We extend our gratitude once again to the academic editor and the two reviewers for their valuable feedback. Under your expert guidance, our manuscript has achieved greater balance and scientific rigor. We have revised the manuscript accordingly based on your additional suggestions.

Below is a detailed point-by-point response.

Response to Reviewer #1:

Comment 1:

Abstract:

(1)Line 13 – suggest changing “Compared with the Complex Training methods that have significant and widespread effects on enhancing explosive power…” to “While evidence have indicated complex contrast training can enhance strength and power…”

Response:

(1)The changes have been made as suggested. (Lines 13-14)

Comment 2:

Introduction:

(1)Line 50 – suggest changing “self-weight” to “bodyweight jumps” and “reduced load” to “assisted jumps”

(2)Line 60 – should be stretch-shortening cycle (not stretch-short).

Response:

(1)Yes, the revised content is more appropriate. We have made the necessary changes.(Lines 50-51)

(2)Thank you for your careful review. This was a spelling error, and we have made the necessary correction. (Lines 60)

Comment 3:

Methods:

(1)Line 117 – the sessions were separated by closer to 72 hours minimum?

(2)Line 235 – typo of after the Tukey method (..)

Response:

(1)After careful inspection and verification, this was found to be a clerical error and has been corrected based on the findings. “closer to 72 hours minimum”. (Lines 117)

(2)Already modified. (lines 235)

Comment 4:

Discussion:

(1)Line 328 – I would suggest revising the final two sentences of this paragraph. You didn’t measure fatigue, you measured perceived exertion, which are quite different constructs.

(2)Line 333 – typo here (,,)

Response:

(1)Thank you for raising such a detailed question. Indeed, there is a fundamental distinction between “perceived exertion” and “measure fatigue,” and our previous wording placed greater emphasis on “measure fatigue.” However, our revised text highlights “perceived exertion,” making the statement more rigorous. (lines 327-330)

(2)Already modified. (lines 332)

Comment 5:

Conclusion:

(1)Line 406 – it would be good to note that it was better over a short time frame, as the study was only 8 weeks (rather than confidently saying it is simply better).

(2)The link you have shared does not work. Please correct to provide raw data.

(3)Table 1 – need to capitalise the words “isometric” and “back”.

(4)Table 3, 4, and 5 – suggest changing “MS” to “1RM Squat”

Response:

(1)Indeed, the phrasing “more effective in a shorter period of time” is more precise. We have made the corresponding revisions in the Conclusion and Abstract sections. (lines 28) (lines 407)

(2)We have examined the shared link and confirmed that it was indeed experiencing issues, though the exact cause remains unclear. We have since updated the link and verified that it is now accessible. Here is the updated link: https://doi.org/10.6084/m9.figshare.30349558.v1. (lines 419)

(3)Already modified. (Table 1)

(4)Already modified. (lines 278-279) (Table 3, 4, and 5)

Response to Reviewer #2:

Comment 1:

Introduction:

(1)Page 18, Line 348: The phrase "strong muscle contractions, as well as effective utilization of the stretch-shorten cycle" is clear. However, the track-changes document indicates the authors intended to remove "centripetal and centrifugal," as agreed upon in their response to Reviewer #1 (Point 7) , but this phrase was not fully removed and remains in the clean version. Please remove it as agreed.

Response:

(1)We have reviewed the entire text and found that lines 318-320 appear to contain relevant descriptions. We have removed the expressions in this section.

Comment 2:

Methods:

(1)Page 12, Line 221 (and multiple other locations): There is a significant contradiction regarding data availability. The "Response to Reviewers" letter states the data is on Figshare (DOI: 10.6084/m9.figshare.30349558). The "Date Availability Statements" section in the main manuscript (Page 42, Line 988) also lists this Figshare DOI. However, the "Data Availability" section of the submission form (Page 12, Line 221) states: "All the data obtained from the experiments can be retrieved from the ScienceDB database (dataset number: 10.57760/sciencedb. 18250)". Please resolve this contradiction and ensure the correct repository and DOI/accession number are stated consistently throughout all parts of the submission.

Response:

(1)We have now ensured that the correct repository and DOI/accession number are consistently stated in all submitted sections. Our links have been updated in the manuscript. Here is the updated link: https://doi.org/10.6084/m9.figshare.30349558.v1.

Comment 3:

Discussion:

(1)Page 40, Line 956: The "Practical Applications" section recommends "isometric back squat". This is inconsistent with the term used in the Conclusion. Please ensure consistent terminology.

Response:

(1)In the latest version, we have standardized the term “isometric back squat” and ensured consistent terminology.

Comment 4:

Conclusion:

(1)Page 16, Line 289 (from Abstract): There is a typographical error: "The findings suggest that The French Contrast..." This should be corrected to "The findings suggest that the French Contrast...".

(2)Page 42, Line 972 / Page 101, Line 2366: The authors correctly adopted the suggestion to change "can" to "may" ("These advantages may be attributed..."). However, the conclusion (Page 40, Line 956) and "Practical Applications" (Page 101, Line 2362) use the term "isometric static deep-back squat." This terminology is not used in the Methods section or Table 1, which use "isometric back squat." Please standardize this terminology throughout the manuscript for clarity.

Response:

(1)Thank you very much for your thorough review. This was a printing error, and we have already corrected it. (lines 26)

(2)After our verification, the term “isometric back squat” has been consistently used throughout the entire paper.

We would like to express our sincere gratitude once again to the editors and reviewers for their diligent work. Your valuable suggestions have significantly enhanced the quality of our manuscript. We hope the revised version meets your expectations and look forward to your favorable consideration. Should you have any questions, please do not hesitate to contact me.

Sincerely,

Yuhua Gao, PhD

Guangzhou sport University, China

Email: hry13066290079@gmail.com

---

## [Editor Report · Decision Letter 2]

24 Nov 2025

Comparative Effects of French Contrast Method vs. Complex Training on Explosive Power and its Endurance in Youth Badminton Athletes

PONE-D-25-41210R2

Dear Dr. Gao,

We’re pleased to inform you that your manuscript has been judged scientifically suitable for publication and will be formally accepted for publication once it meets all outstanding technical requirements.

Kind regards,

Javier Abián-Vicén, Ph.D.

Academic Editor

PLOS ONE

Additional Editor Comments (optional):

I am pleased to inform you that your manuscript, “Comparative Effects of French Contrast Method vs. Complex Training on Explosive Power and its Endurance in Youth Badminton Athletes,” has been accepted for publication. You have responded comprehensively to the reviewers’ comments, and your revisions have strengthened the clarity and scientific rigor of the work. As both reviewers were satisfied with the changes and no further concerns remain, we are happy to proceed with the article in its current form. Congratulations on this achievement.
---

## [Editor Report · Acceptance letter]

PONE-D-25-41210R2

PLOS ONE

Dear Dr. Gao,

I'm pleased to inform you that your manuscript has been deemed suitable for publication in PLOS ONE. Congratulations! Your manuscript is now being handed over to our production team.

Kind regards,

on behalf of

Dr. Javier Abián-Vicén

Academic Editor

PLOS ONE